# Morphological and Molecular Functional Evidence of the Pharyngeal Sac in the Digestive Tract of Silver Pomfret, *Pampus argenteus*

**DOI:** 10.3390/ijms24021663

**Published:** 2023-01-14

**Authors:** Huan Jiang, Jiabao Hu, Huihui Xie, Man Zhang, Chunyang Guo, Youyi Zhang, Yaya Li, Cheng Zhang, Shanliang Xu, Danli Wang, Xiaojun Yan, Yajun Wang, Xubo Wang

**Affiliations:** School of Marine Sciences, Ningbo University, Ningbo 315211, China

**Keywords:** *Pampus argenteus*, pharyngeal sac, morphology, papilla, transcriptome

## Abstract

The pharyngeal sac is a comparatively rare organ in the digestive tract among teleost fishes. However, our understanding of this remarkable organ in the silver pomfret (*Pampus argenteus*) is limited. In the present study, we examined the various morphological and histological characteristics of the pharyngeal sac using histochemical techniques and electron microscopy. The pharyngeal sac showed unique characteristics such as well-developed muscular walls, weakly keratinized epithelium, numerous goblet cells, and needle-like processes on the papillae. The porous cavity of the papillae contained numerous adipocytes and was tightly enveloped by type I collagen fibers. These structures might provide mechanical protection and excellent biomechanical properties for grinding and shredding prey. A comparison of gene expression levels between the pharyngeal sac and esophagus using RNA-seq showed that phenotype-associated genes (epithelial genes and muscle genes) were upregulated, whereas genes related to nutrient digestion and absorption were downregulated in the pharyngeal sac. These results support the role of the pharyngeal sac in shredding and predigesting food. Overall, these findings provide a clearer understanding of the pharyngeal sac morphology and explain the morphological adaptations of the digestive tract for feeding on gelatinous prey. To our knowledge, this is the first report on pharyngeal sac gene expression in *P. argenteus*.

## 1. Introduction

The digestive systems of vertebrates have the classic function of transforming complex, non-absorbable food into simple, absorbable forms [1]. According to their distinct feeding habits, diets, and environments, fishes exhibit specific morphological characteristics of the digestive tract and specialized functions necessary to obtain food and survive [2,3,4,5]. Notably, most stromateoidei fishes (Teleostei: Perciformes) are well characterized by a pharyngeal sac in the digestive tract and have an unusual diet of medusae during part or all of their lives [6,7]. The pharyngeal sac is a saccular outgrowth located on the proximal portion of the esophagus, with internally muscular walls covered by numerous projections commonly referred to as papillae [6,8,9,10]. Early research on the pharyngeal sac focused on its morphology and the morphological characteristics that can be used as taxonomic characters [11,12,13]. The pharyngeal sac was initially described as an extra-esophageal growth or pharyngeal origin [7,8]. However, recent research analyzing its skeletal and muscular components has suggested that this organ originated from both the gill arches and the esophagus [14]. In addition, previous studies have examined the morphology, histology, and digestive enzyme activity of the pharyngeal sac [15,16,17]. However, to our knowledge, no detailed studies about the gross anatomy and histology of the pharyngeal sac and papillae are available, and related molecular research on the pharyngeal sac has not been reported to date.

The silver pomfret (*Pampus argenteus*), a member of the suborder stromateoidei, is characterized by the presence of a pharyngeal sac in the digestive tract [6,18]. Their diet consists of copepods, other non-copepod crustaceans, molluscs, ctenophores, medusae, and fish larva [19,20,21]. As an economically important marine fish worldwide, the artificial propagation and selective breeding of *P. argenteus* have been studied extensively. The food items used for farmed *P. argenteus* mainly comprise copepods, jellyfish, mackerel surimi, and Yubao compound feed [22]. However, owing to a lack of information on the basic biology of this fish, on some occasions, poor nutrition leads to slow growth of *P. argenteus* during breeding. It is known that the structural adaptations in the digestive tract of fishes can be used to understand why and how particular fish species eat and use unique food resources. Studies on catfish (*Pachypterus khavalchor*) reported that the structure of digestive tract was related to a scale-eating habit [5]. In addition, the structure and function of the digestive tract in the Gangetic mud-eel (*Ophichthys cuchia*) supported its utilization of mud-dwelling prey [1]. Thus, it is necessary to investigate the morphofunctional characteristics of the pharyngeal sac to provide a theoretical basis for *P. argenteus* farming.

So far, the morphological and histochemical characteristics of the digestive tract have been studied in many marine and freshwater fishes [23,24,25,26], and transcriptomics has become a powerful tool in fish studies for understanding complex interactions between phenotype and genotype [27]. For example, RNA-seq was performed on farmed *Takifugu rubripes* to study the differential gene expressions between the intestine and stomach, and the key genes (fatty acid binding protein 6, *FABP6*; apolipoprotein A, *APOA*; cadherin 2, *CDH2*; and claudin 3a, *CLDN3A*) and pathways related to metabolism and digestion were identified [28]. Moreover, certain genes (trypsinogen, pepsinogen, and cathepsin) encoding proteolytic enzymes necessary for digestion and absorption in the yellowtail (*Seriola quinqueradiata*) gastrointestinal tract were determined using RNA-seq [29]. The intestinal tract of the European sea bass (*Dicentrarchus labrax*) has been studied as well; the representative genes were involved in intestinal permeability, macronutrient and micronutrient digestion and absorption, immune response, and intestinal chemosensing [30]. In the present study, we used various histochemical techniques including hematoxylin–eosin, Masson’s trichrome, Sirius red, and oil red O staining and electron microscopy, scanning electron microscopy (SEM), and transmission electron microscopy (TEM) to describe the structure of the pharyngeal sac and the papillae upon it. A comparative transcriptomic analysis between the pharyngeal sac and esophagus was performed to identify the gene governing the phenotype and function divergence. Our results are expected to provide further basic biological knowledge about the pharyngeal sac, which is not only beneficial to *P. argenteus* farming, but also serves as a reference for future research on this unique organ.

## 2. Results

### 2.1. Gross Anatomy of the Pharyngeal Sac

It was observed that the digestive tract of *P. argenteus* consists of the oropharyngeal cavity, pharyngeal sac, esophagus, stomach, pyloric caeca, and long intestine, with a total length of 45.12 ± 0.27 cm (Figure 1A). The pharyngeal sac was located at the vertebral column on the dorsal side of the body, opening and connecting to the last gill arch, followed by a short esophagus. The pharyngeal sac was divided into two bilateral compartments by a sagittal sulcus (muscle ridge), with a mean size 6.3 ± 0.27 cm and a mean length 2.3 ± 0.5 cm. The pharyngeal sac was brown-black, with a muscular external appearance (Figure 1B,C). The inner muscular wall of the pharyngeal sac was coated with sticky mucous secretions and showed numerous, hard papillae with needle-like secondary processes. The longest papillae were located in the center of the sac; from the center outward, the number of papillae increased, but their height gradually reduced (Figure 1D). The papilla had an elongated axis and a bottom with a 5–8 radial cone-shaped stellate base, which was anchored in the muscle layer to support the papilla (Figure 1E).

### 2.2. Analysis of Key Morphological Characteristics of the Pharyngeal Sac

The pharyngeal sac consisted of four layers—mucosa, submucosa, muscularis, and serosa—from internal to external (Figure 2A). The serosa was composed of connective tissue, melanocytes, and blood vessels (Figure 2B,C). Next, we focused on the distinctive morphological features of the pharyngeal sac.

#### 2.2.1. Muscularis

Histologically, the muscularis of the pharyngeal sac was well developed and composed of striated muscle. There was no obvious boundary between the two muscle layers, namely, the outer longitudinal layer and the inner circular layer (Figure 3A,B). TEM showed a small amount of intercellular matrix and loose connective tissue between muscle fiber bundles (Figure 3C,D). The sarcomere length was about 3.5 μm, with the A band in the bright area and the I band and the Z band in the dark area (Figure 3E). In addition, a muscular crest was observed in the middle of the pharyngeal sac (Figure 4A). Histologically, the muscular crest mainly consisted of striated muscle, collagen fibers, and blood vessels. The short papillae and stellate bases were visible in the muscular crest. (Figure 4B). With Masson’s trichrome staining, the muscle fiber bundles of the muscular crest were stained red, and the collagen fibers were stained blue (Figure 4C,D).

#### 2.2.2. Papilla

The papillae were located in the mucosa, submucosa, and muscularis layers. Through the full view of the papilla, three well-defined regions were observed: the elongated axis, secondary processes, and stellate base. For the first region, the elongated axis of the papilla was composed of the epithelium layer, collagen fiber layer, and adipose tissue layer from outside to inside. The epithelial layer contained stratified squamous cells at the top of papillae, interspersed with goblet cells. The goblet cells showed mucus-filled cytoplasm and were mainly found in the basal region of the papilla (Figure 5A). Under the SEM, oval or circular columnar epithelial cells appeared embossed and contained densely packed, stubby microridges on the surface of the papilla. In some regions, the epithelial cells were disjointed by the opening of mucous cells (Figure 5B,C). TEM showed the presence of microvilli on the surface of epithelial cells, whereas goblet cells were observed to be filled with mucus granules, vesicles, and electron-dense bodies (Figure 5D). In addition, the collagen fiber layer was composed of type I collagen fibers arranged in parallel to form a close-knit envelope, separating the epithelial layer from the adipose tissue layer (Figure 5H–K). A longitudinal section of the papilla showed the collagen fiber layer and the internal reticular porous structure (Figure 6C). Moreover, many fat cells, blood vessels, and loose connective tissues were observed in the internal structure of the papillae (Figure 5A). Oil red O staining showed that lipid droplets were distributed abundantly in the porous cavity of the papilla (Figure 6D,E). For the second region, a cross section of the papilla showed eight secondary processes symmetrically arranged around the papilla, the bottom was connected to the papilla, and the tip was solid (Figure 6A,B). SEM showed that the tips of the secondary processes penetrated through the mucosal epithelium and were exposed to the pharyngeal sac cavity with their non-cellular and smooth surface (Figure 5E,F). For the third region, the stellate bases were wrapped by epithelial cells and anchored in a staggered manner in the submucosa and muscularis layers, with spaces between them (Figure 5A). The SEM results showed the internal structure of the stellate base (Figure 6G,H). TEM and Masson’s trichrome staining results showed that it was composed of mineralized chitin arranged in parallel and collagen fibers (Figure 6I). On the basis of these findings, an illustration was created to depict the structure of the papillae (Figure 7).

### 2.3. De Novo Assembly and Functional Annotation

A total of six libraries were constructed from the pharyngeal sac and the esophagus. The clean reads of each sample ranged from 19.89 million to 23.78 million, Q30 was ≥92%, and GC content was ≥46% (Table 1).

The mapping rate for the large yellow croaker (*Larimichthys crocea*) genome was approximately 71–78% (Appendix A). An optimized assembly of the reads generated 41,683 unigenes and 105,485 transcripts. The unigenes’ length distribution of minimum length, mean length, and N50 length were 301 bp, 1898 bp, and 2910 bp, respectively (Appendix A). In addition, the BUSCO analysis results yielded 978 BUSCOs, including 926 (94.6%) complete BUSCOs, 31 (3.16%) fragmented BUSCOs, and 21 (2.14%) missing BUSCOs (Appendix A). The Pearson correlation analysis results among samples showed that R^2^ values ranged from 0.68 to 0.75, indicating the difference between the two groups was small (Appendix A). Raw read data were uploaded to the NCBI Sequence Read Archive (SRA) under the accession number PRJNA889559.

Among the 41,683 unigenes, 33,654 (80.73%) were annotated in the Nt database, and only 10,032 (24.06%) were annotated in the KOG database (Table 1). These transcripts were also tested against the NR database, and the species distribution showed that *Lates calcarifer* accounted for 19.3% of all homologous genes, followed by *Seriola dumeri* (12.2%) (Appendix A). The Venn diagrams regarding annotation in five databases displayed that 8485 unigenes were annotated in all databases, while 9361 unigenes were uniquely annotated in the Nt database (Appendix A). All these results indicated that the sequences were appropriate for further analysis.

All unigenes were classified into three major functional categories. There were 42 significant GO categories, including biological process (25 subclasses), cellular component (5 subclasses), and molecular function (12 subclasses). The GO category of biological processes showed a high percentage of biological processes. In terms of biological processes, “cellular process”, “metabolic process, “regulation of biological process”, “biological regulation”, and “response to stimulus” were highly enriched. The cellular component category contained “cellular anatomical entity” and “intracellular and protein containing complex” subclasses. With regard to molecular function, the top subclasses were “binding” and “catalytic activity” (Appendix A). In addition, in the functional classification of the KEGG pathways, there were 291 different pathways classified into five main categories, involving metabolism, environmental information processing, cellular processes, genetic information processing, and organismal systems. Among them, metabolism contained 127 sublevel pathways, including “carbohydrate metabolism”, “lipid metabolism”, “amino acid metabolism”, etc. Other representative pathways enriched many unigenes such as “signal transduction” (2509 unigenes), “immune system” (1220 unigenes), and “endocrine system” (1122 unigenes) (Appendix A). In addition, 10,032 (24 %) unigenes were annotated in KOG and grouped into 26 classifications. The largest cluster was “signal transduction mechanisms” (1842 unigenes), followed by “general function prediction only” (1586 unigenes) and “posttranslational modification”, “protein turnover, chaperones” (999 unigenes) (Appendix A). The top transcription factors in the pharyngeal sac were compiled and listed in Appendix A.

### 2.4. DEGs Potentially Correlated with Structure and Function of the Pharyngeal Sac

The DEGs between pharyngeal sac and esophagus were analyzed to identify tissue-specific gene expression patterns. The heatmap results showed that there was a large number of genes with different expression patterns between the pharyngeal sac and the esophagus (Figure 8A). The padj value < 0.05 and |log_2_Fold Change| > 1 were used as the threshold for detecting significant DEGs. A total of 8538 unigenes were identified as DEGs, including 4311 upregulated unigenes and 4227 downregulated unigenes (Figure 8B). DEGs were grouped into four categories associated with enzyme genes (e.g., *PGA*, *AMSase*, *SLC6A18*, *PRSS2*, and *CRTB*), muscle-related genes (e.g., *MYH*, *MYLPF*, *COL1A*, and *COL10A1*), immune-related genes (e.g., *MRC*, *TOLLIP*, and *IGHM*), and metamorphosis-related genes (e.g., *WNT5*, *TBX4*, and *PAX9*) (Appendix A).

According to the GO functional analysis, 10,991 DEGs were grouped into the biological process category, 9326 DEGs into the cellular component category, and 6127 into the molecular function category. The “structural constituent of ribosome” (GO:0003735) was the most significantly enriched GO term (Figure 9A). The GO terms of the upregulated DEGs were mainly about muscle-related terms such as “cell adhesion” (GO:0007155), “cytoskeleton” (GO:0005856), and “ion binding” (GO:0043167) (Figure 9B). The downregulated GO terms were primarily concerned with “small molecule metabolic transport” (GO:0044281), “cytoplasm” (GO:0005737), and “transmembrane transporter activity” (GO:0022857) (Figure 9C).

KEGG enrichment analysis of the DEGs identified 341 signaling pathways, of which 12 were significantly altered. The top three pathways enriched were “oxidative phosphorylation” (ko00190), “Alzheimer disease” (ko05010), and “Parkinson disease” (ko05012), suggesting the critical role of energy metabolism and immune systems in the pharyngeal sac. Many DEGs were involved in “focal adhesion” (ko04510), “tight junction” (ko04530), “regulation of actin cytoskeleton” (ko04810), and “cardiac muscle contraction” (ko04260), which were essential for epithelial morphology and muscle contraction. The “protein digestion and absorption” (ko04974) pathway was also enriched (Figure 10).

### 2.5. Validation of Gene Expression Levels

To validate the expression levels of the DEGs obtained from the comparative RNA-seq analysis, 26 DEGs from various categories were selected for confirmation. Figure 11 showed that that the RT-qPCR data were basically consistent with the RNA-seq analysis results, supporting the reliability and accuracy of the RNA-seq analysis (Figure 11).

## 3. Discussion

### 3.1. Morphology of the Pharyngeal Sac

In the present study, we provided a clear description of the morphological and histological features of the pharyngeal sac in *P. argenteus*. We found that the pharyngeal sac is composed of four tissue layers—the mucosa, submucosa, muscularis, and serosa—which is consistent with the previous descriptions of the digestive tract in vertebrates [25,31,32]. This study examined certain prominent features of the pharyngeal sac. First, the muscularis of the pharyngeal sac was particularly well developed, implying that food could be controlled to pass through the sac. Another prominent feature is the papillae lining the lateral walls; these papillae structures have been referred to as esophageal or pharyngeal sac “teeth” in previous studies [7,11,13], and were considered to be composed of homogeneous calcified tissue, pulp tissue, or bone tissue [11,33]. Contrary to these previous findings, we found that the papillae were composed of an epithelium layer, collagen fiber layer, and adipose tissue layer. The epithelium layer may protect the papilla from mechanical and chemical abrasions, as the surface of the papilla was most extensively exposed to friction during feeding. Moreover, the papillae had an internal honeycomb structure and secondary processes, which may make them lightweight and facilitate their ability to pierce food. Another key feature of the papillae was the stellate base anchored in the different muscle layers to improve space utilization. Although direct observation of the feeding behavior and mechanics was not feasible in the present study, the presence of features such as well-developed muscular walls; weakly keratinized epithelium cells; numerous goblet cells; and needle-like processes on the papillae confirmed the biomechanical role of papillae in the grinding and shredding of prey. Moreover, these findings suggest that the presence of the pharyngeal sac compensates for the small mouth and underdeveloped teeth in *P. argenteus* [34]. Similar papillae with finger-like or polypoid morphology have been reported in the masked angelfish (*Genicanthus personatus*). However, histological findings suggested that they contained no blood vessels or capillaries to absorb nutrients, which is inconsistent with our observations in *P. argenteus* [35]. Likewise, studies on the yellowtail fusilier (*Caesio cuning*) reported that forked papillae were located in the center of the esophagus and pointed backward to keep food moving down the esophagus [36]. Moreover, keratinized oropharynx papillae have been reported in turtles feeding on jellyfish; these papillae aid in shredding and provide mechanical protection from dangerous prey such as jellyfish [37]. The existence of these distinctive structure in different species, including *P. argenteus,* signifies adaptations related to food habits, dietary specialization, and effective manipulation of food. An important direction for future research on this specialized digestive tract is the underlying evolutionary mechanism.

The presence of papillae has a physiological significance likely associated with prey selection. In the natural environment, *P. argenteus* feed on gelatinous organisms such as medusae, ctenophores, and siphonophores [38], indicating that this fish possesses some immunity or protection mechanism against toxins. Previous studies have reported that the pharyngeal sac has a protective mucus-secreting system against the toxins of medusae [9]. In the present study, a novel finding was that the porous cavity of the papillae was filled with abundant adipocytes and enveloped by tightly arranged collagen fibers. However, few studies have been reported about the abundant adipocytes found in the digestive tract. It has been reported that large adipocytes typically show high metabolic activity and secrete higher quantities of immune cell chemoattractants [39]. Most stromateoidei fish have been reported to have an intradermal canal plexus situated under the epidermis, which is distributed all over the body and filled with a viscid, oily substance [9]; a recent report indicates that this system potentially protects the fish against jellyfish stings [14]. Inspired by this report, we speculate that the large amount of lipids accumulated in the porous cavity of the pharyngeal sac papillae is associated with feeding on and detoxification of jellyfish. Further research is required to identify the role of adipocytes in the papillae.

### 3.2. Candidate Genes Determined from Correlation between Structure and Function

To date, comparative transcriptome analyses have been performed in teleost species under normal physiological conditions. For example, RNA-seq analysis of three major organs (stomach, intestine, and rectum) was performed in *S. quinqueradiata* to understand digestion and nutrient absorption at the molecular level [29]. Moreover, the functional and transcriptional compartmentalization of the digestive tract in zebrafish (*Danio rerio*) has also been studied [40]. In the present study, we compared the gene expression levels between the pharyngeal sac and esophagus in *P. argenteus* by RNA-seq to identify tissue-specific gene expression patterns. The following mainly discusses the genes determined on the basis of correlation between structure and function.

Based on the RNA-seq data, we identified the candidate genes associated with complex phenotypes, such as epithelial genes and muscle growth genes. As keratin is a key part of the epithelial cytoskeleton, the keratin gene family (*KRT*) is important for the mechanical stability and integrity of epithelial cells and tissues. Fibrous proteins of this type form the structural framework of cells and constitute the tough skin, hair, and nails [41]. In the present study, the differentially expressed genes *KRT8* and *KRT13* were significantly enriched in the pharyngeal sac. Previous research has shown that *KRT8* is expressed in the epithelial cells of the ectoderm and periderm in human fetal skin [41,42]. Moreover, *KRT13* is expressed in the oral mucosa, esophagus, and forestomach, as the characteristic acidic keratin is produced in the superbase cells of non-cornified stratified epithelia [43]. These results suggested *KRT8* and *KRT13* play a role in forming the stratified epithelia of the pharyngeal sac. Similarly, muscle growth-related genes (e.g., *MYL*, *MYH*, *MYLK*, and *COL1A*) were significantly enriched in the pharyngeal sac. *MYH* and *MYL* have been reported as typical regulators of muscle contraction and are expressed in striated muscles [44,45]. To some degree, the high expression levels of muscle growth-related genes may play an important role in the muscle growth and motility of the pharyngeal sac.

As expected, our result showed that genes encoding proteolytic digestive enzymes (e.g., *PGA*, *PRSS2*, and *AMCase*) and lipid metabolism-related genes (e.g., *APOE*, *LRP1*, and *FASN*) were enriched significantly. Previous studies reported that *PGA* is involved in breaking down long-chain peptides during the earliest stage of protein digestion in the stomach of *S. quinqueradiata* [29]. *PRSS* plays a key role in hydrolyzing proteins and activating other digestive zymogenes [46]. *AMCase* exhibited chitinase activity in the esophagus, which was associated with the digestion of chitinous substances from crustaceans [47]. However, our results showed that the above enzyme-encoding genes were downregulated in the pharyngeal sac, indicating that the esophagus has a stronger digestive function than the pharyngeal sac. This finding is consistent with our previous finding that the pharyngeal sac exhibits the lowest digestive enzyme activity [17]. Moreover, the functions of the digestive tract include not only digestion and nutrient absorption but also immune defense against pathogens [48]. The intracellular TIR domain of Toll-like receptors (*TLRs*) interacts with adaptor proteins and triggers a signaling cascade that ultimately induces an immune response in the body [48]. *TOLLIP* encodes a key negative regulator of TLR-mediated innate immune responses [49]. These immune genes (e.g., *MRC*, *TOLLIP*, *TLR3*, and *IGHM*) were highly expressed in the pharyngeal sac. Moreover, the KEGG pathway related to human diseases was also significantly enriched. Although the role of these genes in the pharyngeal sac remains unclear, the present findings indicate that the pharyngeal sac, as the first segment of the digestive tract, has its own defense mechanism to protect against pathogens and toxins. Further research is needed to determine whether the pharyngeal sac has an immunological role in the digestive tract.

In conclusion, our findings provide a clearer understanding of the morphological and histological features of the pharyngeal sac in *P. argenteus*, confirming that its structure has an important biomechanical function in the grinding and shredding of prey. Moreover, to the best of our knowledge, this is the first report on gene expression in the pharyngeal sac. Candidate genes and KEGG pathways were associated with phenotype (epithelial and muscle genes), digestion, nutrient absorption, and immune function. Overall, the study of the pharyngeal sac is still a developing field of research with many knowledge gaps. Our results are expected to serve as a reference for future research aimed at studying the pharyngeal sac.

## 4. Materials and Methods

### 4.1. Experimental Fish and Sampling

Thirty adult individuals of *P. argenteus* (mean body weight 35.53  ±  0.32 g, mean body length 14.65 ± 2.16 cm) were obtained from Xiangshan Harbor Aquaculture and Larva Company (Ningbo, China). Healthy fish were cultivated in breeding ponds (temperature = 22.0−24.0 °C, dissolved oxygen = 9.85 mg/L, salinity = 33%, pH = 8.56). The fish were fed three times a day with commercial feed (Larvae Love, #6, Tokyo, Japan) equivalent to 2–3% of their body weight and were starved for 8 h before the experiment. Then, fish were euthanized using 75 mg/L buffered MS-222 (Finquel MS-222, Sigma Inc., Marlborough, MA, USA), and the pharyngeal sac and the esophagus were dissected, incised longitudinally, and spread out (*n* = 3). The samples were washed with 0.1 M phosphate-buffered saline and divided into four parts (Solarbio, Beijing, China): one part was fixed in 4% paraformaldehyde (Beyotime, Shanghai, China) to be used for paraffin sections; one part was embedded in optimal cutting temperature (OCT) compound to be used for frozen sections; a third part was placed in 2.5% glutaraldehyde solution (Innochem, Beijing, China) for electron microscopy observations; and for the fourth part, samples from three fishes were pooled as one sample and then rapidly preserved at −80 °C for RNA-seq and quantitative real-time PCR (RT-qPCR).

### 4.2. Light Miroscopy

The gross anatomy of the digestive tract was examined using a stereomicroscope (Leica, Germany), and photographs were taken in situ. Histological experiments were performed as previously described [50]. In brief, the samples were fixed in 4% paraformaldehyde for more than 48 h, dehydrated in a graded series of ethanol, and then cleared in xylene. Then, they were embedded in paraffin and sectioned into 5 µm thick sections using a table microtome (Leica, Wetzlar, Germany); these sections were stained with hematoxylin and eosin (HE), Masson’s trichrome, Alizarin red, and Sirius red. Samples embedded in OCT were cut into 10 µm thick sections, which were stained with oil red O and stored at 4 °C. All sections were observed and imaged using a positive fluorescence microscope (Nikon, Tokyo, Japan), except for sections with Sirius red staining, which were observed using polarized light microscopy (Nikon, Tokyo, Japan).

### 4.3. Electron Microscopy

For SEM, the fixed pharyngeal sac and papillae were washed in 0.1 M sodium phosphate buffer (Real-Times, Beijing, China); dehydrated in a graded series of tert-butanol (Aladdin, Shanghai, China); and then dried using a critical point dryer (Nikon, Tokyo, Japan) for 24 h, mounted on carbon tape, and coated with gold palladium (approximately 30 nm thick). The samples were observed under a Hitachi S-3400N (Hitachi, Tokyo, Japan) operating at 25 kV.

For TEM, small fragments of the pharyngeal sac and papillae were washed in 0.1 M sodium phosphate buffer and post-fixed in 1% osmium tetroxide for 2 h at 25 °C. The samples were rinsed thoroughly with the same buffer (Real-Times, Beijing, China), dehydrated in an ascending series of acetone (Aladdin, Shanghai, China), and embedded in araldite resin. Then, semithin (60–90 nm thick) sections were cut using an ultramicrotome (Leica EM UC7, Wetzlar, Germany) and mounted on copper grids. Finally, the sections were stained with uranyl acetate and lead citrate, and images were captured under a Hitachi S-7650 (Hitachi, Tokyo, Japan) operating at 200 kV.

### 4.4. RNA Isolation, Library Preparation and Sequencing

Total RNA was extracted in 1.5 mL centrifuge tubes (Jet Biofil, Guangzhou, China) by using FastPure^®^ Cell/Tissue Total RNA Isolation Kit V2 (Vazyme, Nanjing, China) following the manufacturer’s instructions. RNA integrity was assessed on 1% agarose gels and an RNA Nano 6000 Assay Kit of the Bioanalyzer 2100 system (Agilent Technologies, Santa Clara, CA, USA). The mRNA was purified from the total RNA by using poly-T oligo-attached magnetic beads. Fragmentation was carried out using divalent cations under elevated temperature in First-Strand Synthesis Reaction Buffer (5X). First-strand cDNA was synthesized using random hexamer primer and M-MLV Reverse Transcriptase, and then RNase was used to degrade the RNA. Second-strand cDNA synthesis was subsequently performed using DNA polymerase I and dNTP. Remaining overhangs were converted into blunt ends via exonuclease/polymerase activities. After adenylation of 3′ ends of DNA fragments, adaptors with a hairpin loop structure were ligated to prepare for hybridization. In order to select cDNA fragments of preferentially 370~420 bp in length, the library fragments were purified with an AMPure XP system (Beckman Coulter, Beverly, MA, USA). Finally, the Illumina NovaSeq 6000 platform (Beijing, China) was used for sequencing, and the end reads of 150 bp pairing were generated.

### 4.5. De Novo Assembly, Unigene Annotation and Functional Classification

Clean reads were filtered from raw data by removing adapter and low-quality reads. BUSCO analysis was used to evaluate the splicing quality. Then, Pearson correlation analysis was conducted to visualize the relative differences in the six samples. Gene function was annotated based on seven databases: the NCBI non-redundant protein sequences (Nr), NCBI non-redundant nucleotide sequences (Nt), Protein family (Pfam), Clusters of Orthologous Groups of proteins (COG/KOG), a manually annotated and reviewed protein sequence database (Swiss-Prot), KEGG Ortholog database (KO), and Gene Ontology (GO). The GOseq (1.10.0) and KOBAS (v2.0.12) software were used for GO function enrichment analysis and KEGG pathway enrichment analysis. The unigene expression was calculated and normalized with the reads per kb per million reads (RPKM) method (Mortazavi, Williams, Mccue, Schaeffer, & Wold, 2008). The genes with |log_2_ Fold Change| ≥  1 and padj ≤  0.05 were identified as differentially expressed genes by using the DESeq2 R package (1.20.0). The GO function and KEGG pathway enrichment analysis of differentially expressed genes (DEGs) with padj ≤  0.05 were defined as enriched significantly.

### 4.6. Validation of Transcriptome Data by RT-qPCR

To verify the reliability of the transcriptome results, 26 DEGs were selected for validation by RT-qPCR. The Primer 5.0 software was used to design the primers that are shown in Table 2. The system for RT-qPCR contains 5 µL SYBR Green I Master Mix (TransGen Biotech, Beijing, China), 0.2 µL forward and reverse primers, respectively, 1 µL cDNA template, and dd H_2_O added to 10 µL. Each DEG was repeated four times. RT-qPCR was performed on a q225 fluorescence quantitative qPCR machine (Kubo Technology, Beijing, China) with one cycle at 94 °C for 30 s, followed by 40 cycles at 94 °C for 5 s, 59 °C for 15 s, and 72 °C for 10 s. The relative expression levels were normalized to the endogenous control gene *β*-actin [51], and expression ratios were calculated by using the 2^−ΔΔCt^ method [52].

## Figures and Tables

**Figure 1 ijms-24-01663-f001:**
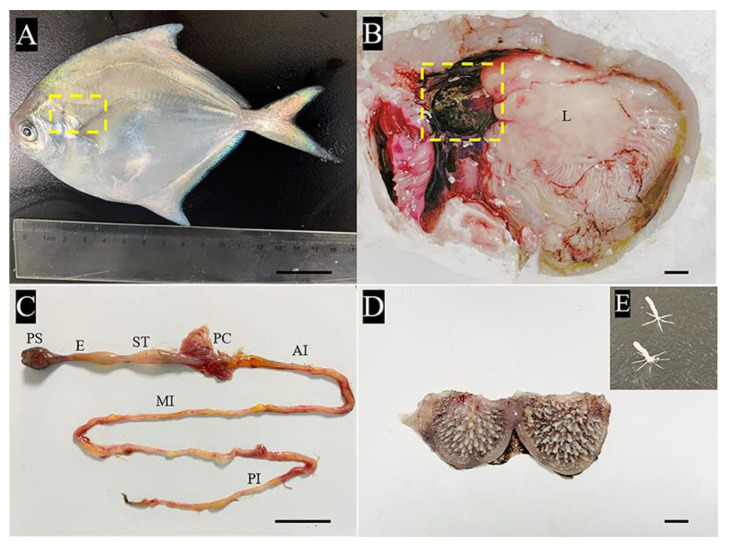
Gross dissection of pharyngeal sac in the silver pomfret (*Pampus argenteus*). (**A**) Ventral views of the external features of the fish used in the present investigation. The yellow box indicates the position of pharyngeal sac. (**B**) Cut open to show internal organs of silver pomfret. The yellow box indicates the position of the pharyngeal sac. Liver (L). (**C**) Structures of the digestive tract: pharyngeal sac (PS), esophagus (E), stomach (ST), pyloric caeca (PC), anterior intestine (AI), middle intestine (MI), and posterior intestine (PI). (**D**) The cut image of the pharyngeal sac, showing the inner surface is covered with numerous papillae distributed over the entire internal surface. (**E**) The papilla, showing the stellate base and elongated axis.

**Figure 2 ijms-24-01663-f002:**
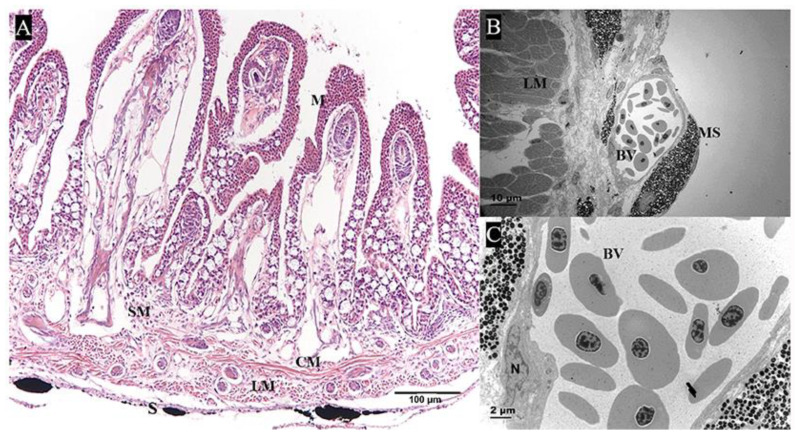
The structure of pharyngeal sac. (**A**) Transverse section of pharyngeal sac, showing the mucosa (M), submucosa (SM), longitudinal muscle (LM), circular muscle (CM), and serosa (S) (Hematoxylin-eosin staining, HE). Bar = 100 μm. (**B**,**C**) The structure of serosa, showing the blood vessel (BV), the longitudinal muscle (LM), and melanocytes (MS) (transmission electron microscopy, TEM), N, nucleus (TEM): (**B**) bar = 10 μm, (**C**) bar = 2 μm.

**Figure 3 ijms-24-01663-f003:**
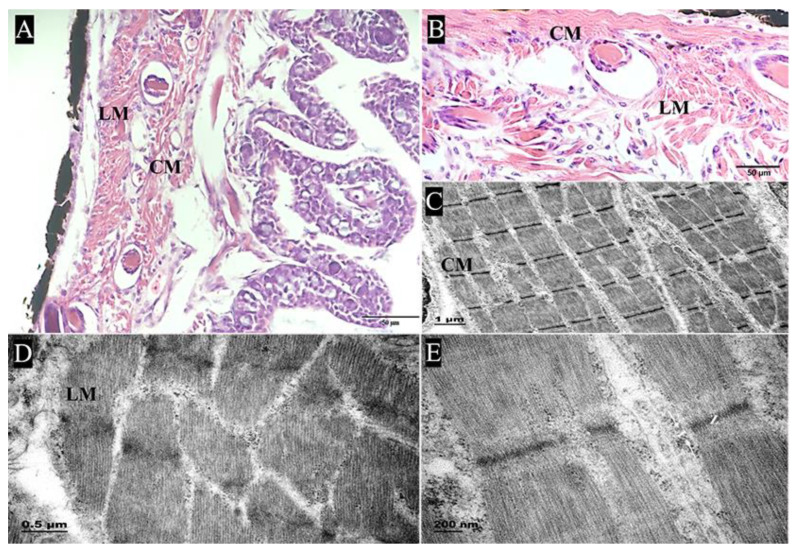
The muscularis structure of the pharyngeal sac. (**A**,**B**) Longitudinal section shows the submucosa and muscular layer, including the outer longitudinal muscle (LM) and inner circular muscle (CM) (HE); bar = 50 μm. (**C**,**D**) Section of muscle fiber bundles, with intercellular matrix and loose connective tissue (TEM); (**C**) bar = 1 μm, (**D**) bar = 0.5 μm. (**E**) Sarcomeres appear as distinct structural units: A band in the bright area, I band in the dark area, and Z band in the middle of the I band (TEM); bar = 200 nm.

**Figure 4 ijms-24-01663-f004:**
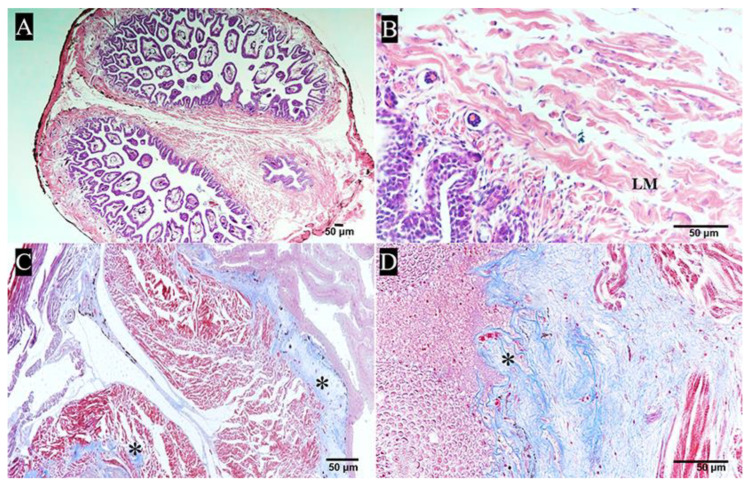
The muscular crest of pharyngeal sac. (**A**,**B**) Central ridge of interior of pharyngeal sac. Longitudinal muscle (LM) (HE); bar = 50 μm. (**C**,**D**) The muscular crest mainly consisted of muscle fiber bundles, collagen fibers, and blood vessels. Collagen fibers are stained blue, while muscle fibers are stained red (Masson’s trichrome staining). * Indicates collagen fibers; bar = 50 μm.

**Figure 5 ijms-24-01663-f005:**
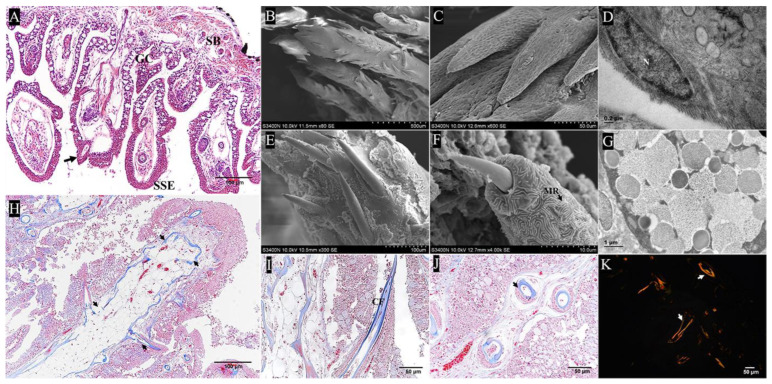
The papilla structure of pharyngeal sac. (**A**) The papilla structure of pharyngeal sac, showing the stratified squamous epithelium cells (SSE), goblet cells (GC), and stellate bases (SB) of primary papilla in muscle layer. The arrow points to needle-like processes of the papilla (HE); bar = 50 μm. (**B**,**C**) The surface structure of the papilla, showing epithelial cells, mucus, and many processes at the top and side of the papilla (scanning electron microscopy, SEM); (**B**) bar = 500 μm, (**C**) bar = 50 μm. (**D**) Epithelial cells with elongated nucleus. N, nucleus (TEM); bar = 0.2 μm. (**E**,**F**) The tips of processes on papilla, penetrating through the mucosal epithelium with fingerprint-like microridges (MR) and exposed to the sac cavity with smooth surface and non-cellular structures (SEM); (**E**) bar = 100 μm, (**F**) bar = 10 μm. (**G**) Goblet cells filled with mucus granules, vesicles, and electron-dense bodies (TEM); bar = 1 μm. (**H**–**J**) The collagen fibers (CF), showing collagen fiber wrapped by mucosal layer in papilla (HE); the arrow points towards collagen fibers. (**H**) bar = 100 μm, (**I**,**J**) bar = 50 μm. (**K**) The type of collagen fiber is type I, stained yellow or red (Sirius red staining). The arrow points towards collagen fibers; bar = 50 μm.

**Figure 6 ijms-24-01663-f006:**
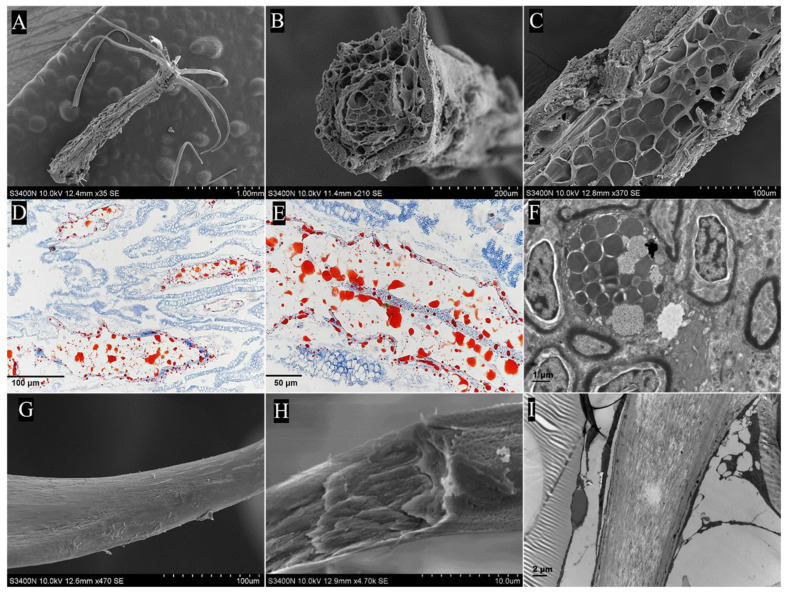
The internal structure of a papilla in the pharyngeal sac. (**A**) Full view of papilla, showing the elongated axis and stellate base (SEM). Bar = 1 mm. (**B**,**C**) Cross and longitudinal sections of papilla, showing long tubular secondary papilla and the internal honeycomb structure (SEM); (**B**) bar = 200 μm, (**C**) bar = 100 μm. (**D**,**E**) Oil droplet distribution in the papilla (Oil red O); (**D**) bar = 100 μm, (**E**) bar = 50 μm. (**F**) Lipid droplet in adipocyte (TEM); bar = 1 μm. (**G**,**H**) Integral and fractured stellate base without epithelial cells (SEM); (**B**) bar = 100 μm, (**C**) bar = 10 μm. (**I**) Interior of stellate base (TEM); bar = 2 μm.

**Figure 7 ijms-24-01663-f007:**
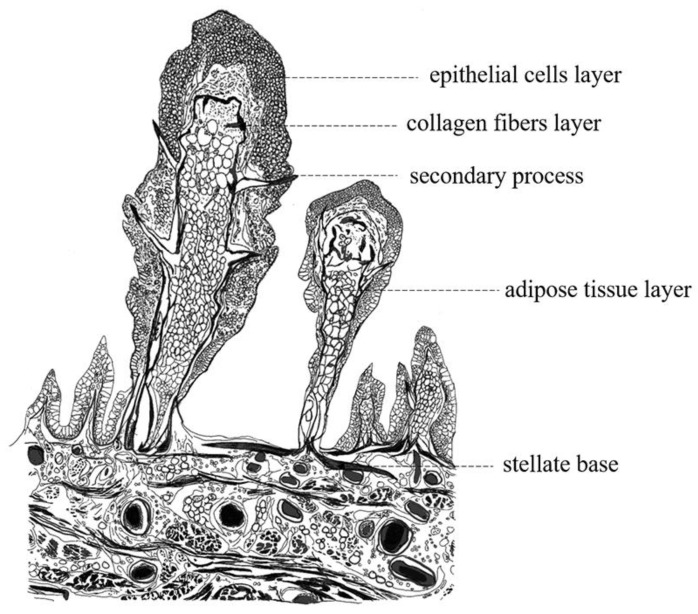
Diagram illustrating structure of pharyngeal sac.

**Figure 8 ijms-24-01663-f008:**
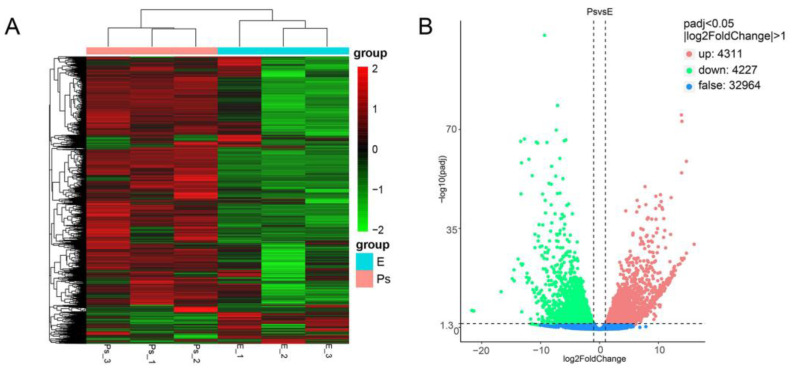
Heatmap and volcano plot and analysis of differentially expressed genes (DEGs). (**A**) Heatmap of DEGs. (**B**) Volcano plot of DEGs. Ps and E indicate the pharyngeal sac and the esophagus, respectively.

**Figure 9 ijms-24-01663-f009:**
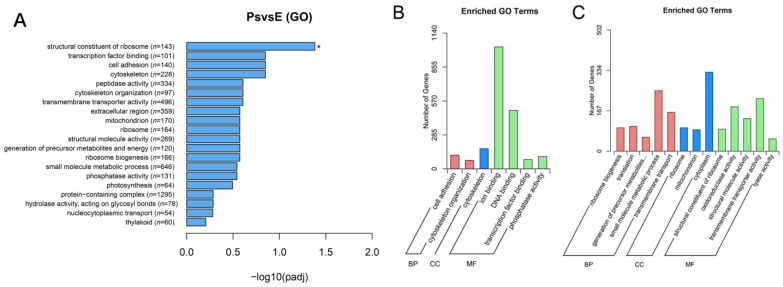
Enrichment and distribution of GO terms of DEGs. (**A**) GO terms of DEGs. * Indicates significantly enriched GO term. Ps and E indicate the pharyngeal sac and the esophagus, respectively. (**B**) The GO terms of the upregulated DEGs. (**C**) The GO terms of the downregulated DEGs.

**Figure 10 ijms-24-01663-f010:**
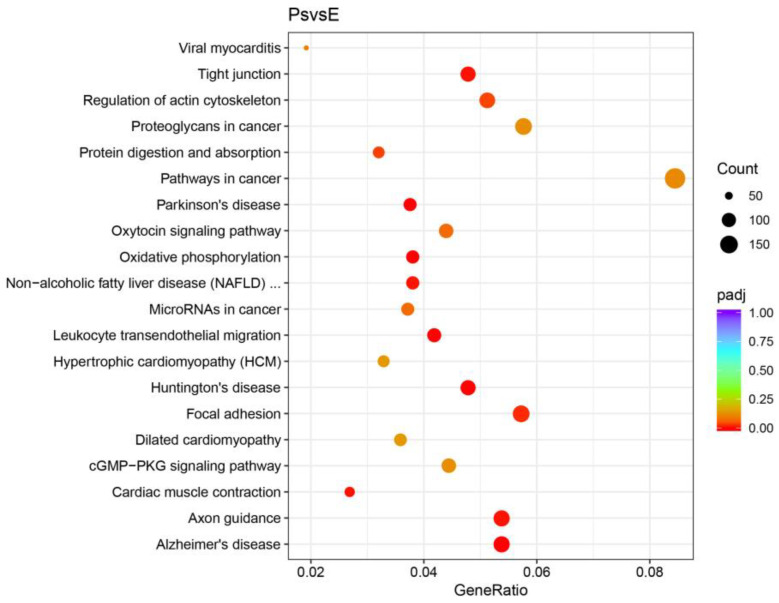
Enrichment of KEGG pathways of DEGs. Ps and E indicate the pharyngeal sac and the esophagus, respectively.

**Figure 11 ijms-24-01663-f011:**
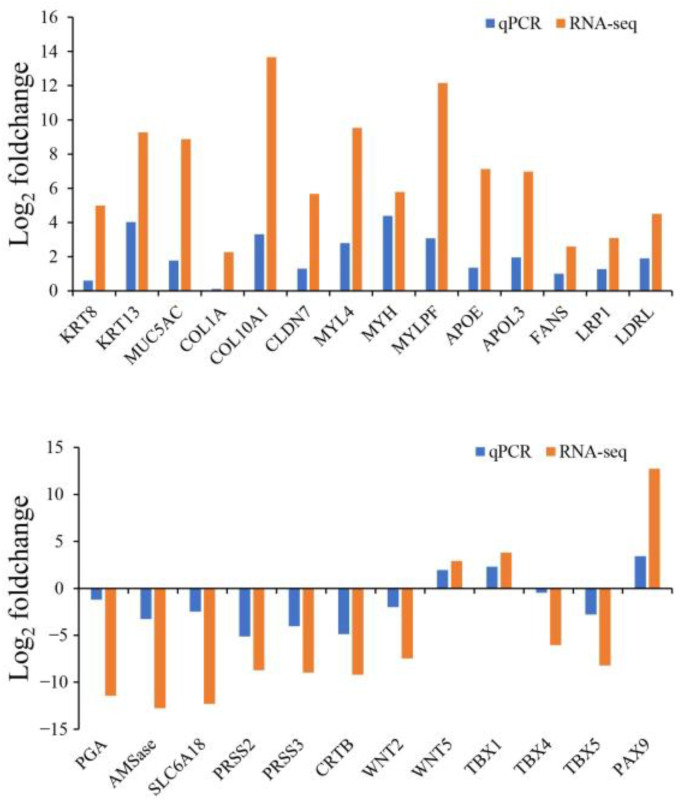
Validation of RNA-seq data by RT-qPCR.

**Table 1 ijms-24-01663-t001:** Statistics of annotation gene number.

Database	Number of Unigenes	Percentage (%)
NR	23,111	55.44
NT	33,654	80.73
KO	14,746	35.37
SwissProt	20,189	48.43
PFAM	18,967	45.5
GO	18,960	45.48
KOG	10,032	24.06
Annotated in all databases	6862	16.46
Annotated in at least one database	34,762	83.39
Total unigenes	41,683	100

**Table 2 ijms-24-01663-t002:** The primers designed for RT-qPCR.

Gene	Gene ID	Primer	Tm (°C)	Primer Efficiency (%)	Product (bp)	Melt Temp.
KRT8	Cluster-5717.6135	F: GATCTGCCGGCTGTAAATGG	58.7	95.2	174	83.69
		R: GTCCTCAAGAATCAGCAGCG	59.2			
KRT13	Cluster-5717.19474	F: ACTTCCTGGAGAGCAAGACC	58.3	99.4	110	85.17
		R: GGCAGCATTTCCACGAGTAG	59.4			
MUC5AC	Cluster-5717.4784	F: TGTGAATGTGTGTGCAGTGG	59.1	101.2	157	81.35
		R: AAGGATCACAGTCATGGGCA	59.7			
COL1A	Cluster-5717.14663	F: GTCCTCTCTCACCAGGCTTT	58.5	85.4	189	90.3
		R: CAAAGGGTATGACTGGCAGC	58.8			
COL10A1b	Cluster-5717.4734	F: TGCAGCCCCTATTCAGTTCA	59.6	89.7	152	83.44
		R: TGTAGAGAGCCACCAATGCA	59.0			
CLDN7	Cluster-5717.4897	F: GCATTCTTGGATCGTTGGCT	58.4	104.6	163	86.48
		R: CAATACCAAGAGCAGGCGAC	59.0			
MYL4	Cluster-5717.4933	F: CACGGGAGATATGTTGCAGC	58.5	96.3	196	85.41
		R: ACCTTGTTTGATCGCACACC	59.6			
MYH	Cluster-5717.4889	F: GGCAGCTTTTGGTTTCTGGA	59.7	95.7	196	84.64
		R: GGGGATTGAGTGGGAGTTCA	59.1			
MYLPF	Cluster-5717.4958	F: AGGCGACCCACATGTTCTTA	58.9	93.6	222	84.69
		R: CCATGATCAAGGAAGCCAGC	59.0			
APOE	Cluster-5717.9257	F: ATGCGGAAGGGAGAACTCAA	58.3	97.4	255	82.55
		R: TCAAGGAGACCACCACTGTC	59.4			
APOL3	Cluster-5717.4775	F: ACTGTCTCCGTTCACTCTGG	59.1	87.5	170	87.83
		R: TGAATCTCAAGCAGGCCTGA	58.6			
FANS	Cluster-5717.20642	F: TTGTTGACCGGCCTATTCCC	58.7	102.1	154	85.99
		R: GGATGCGCCCTAGTCTTCTC	58.9			
LRP1	Cluster-5717.15637	F: CGGGCTGACTGTGTTTGAAA	59.3	86.7	199	87.02
		R: CACCTCACATGCATGGCTAC	59.7			
LDRL	Cluster-5717.25245	F: TGTGAATTCTTGTGCCTGGC	59.0	86.4	171	85.87
		R: ATGAGGTGGGGTGGTAGTTG	58.5			
PGA	Cluster-5717.13230	F: TGGAGGACTGCTTAGGGTTG	58.9	79.4	250	86.32
		R: AAAGGATTGTGGGAGGCGTA	59.3			
AMSase	Cluster-5717.13038	F: AACCGAGTCATCCACCCTTT	58.6	98.6	158	83.54
		R: TGTGTCCCTTCTGCCTGATT	59.2			
SLC6A18	Cluster-5717.11875	F: GTGGGCAGTGCTCTTCTTTC	58.3	97.5	152	85.29
		R: AGAGCCCAACGTGAAGATGA	59.0			
PRSS2	Cluster-5717.10919	F: GGTGGTCCTCTGGTGTGTAA	58.8	93.5	226	86.61
		R: ACCTTCTCCTCCTGTCTCCT	59.3			
PRSS3	Cluster-5717.13926	F: GCAGCAACTATCCTGATCGC	58.9	103.0	233	88.59
		R: GACTCCAGGCCTGTTCTTCT	59.2			
CRTB	Cluster-5717.13085	F: GCACGGACAGCTTAATGAGG	58.7	97.3	196	86.84
		R: ATCAACGAGAACTGGGTGGT	59.0			
WNT2	Cluster-5717.15143	F: TTTGTCCGAAAGTGCTCTGC	58.6	87.5	159	83.9
		R: TCCCAGCAGTTCAGTCTCAG	59.3			
WNT5	Cluster-5717.10855	F: CCTTTGGCAGACTGTTGGAC	58.9	96.4	239	75.08
		R: ACCTTCACCGATGTACTGCA	59.2			
TBX1	Cluster-5717.8690	F: CCCCACCACCTATGAACACT	58.7	85.6	238	86.53
		R: CAGGACTGTGTTGCCTTTCC	59.1			
TBX4	Cluster-5717.11891	F: ACTATGCGAGGAGATGAGCC	58.7	89.7	232	83.41
		R: TGGATCCAAGAACACTGCCT	59.3			
TBX5	Cluster-5717.17018	F: TCCGTCAGAGAGTGTGTTCC	58.9	87.9	218	90.34
		R: CCTCTTGGTGCAGTGGTAGA	59.0			
PAX9	Cluster-5717.24078	F: GTCTACCCCTACAACCCCAC	58.7	78.5	169	83.83
		R: TCATGTTGGTCCAGTCCTCC	59.4			
*β*-acting	[51]	F: ACCCTGAAGTACCCCATCGA	59.1	98.5	157	83.58
		R: GGCCACTCTCAGCTCATTGT	58.9			

## Data Availability

The datasets presented in this study can be found in online repositories. The names of the repository/repositories and accession number(s) can be found below: https://www.ncbi.nlm.nih.gov/ (accessed on 9 January 2023), PRJNA889559.

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
