# Peer review of "Morphological and Molecular Functional Evidence of the Pharyngeal Sac in the Digestive Tract of Silver Pomfret, Pampus argenteus"

_ijms, 2023, doi:10.3390/ijms24021663_

Round 1

Reviewer 1 Report

Revisions for the manuscript with ID (ijms-2058773)

Title:

I suggest changing the title to be

“Morphological and Molecular Functional Evidence of the Pharyngeal sac in the Digestive tract of Silver pomfret, Pampus argenteus

Line 11: “in feeding habits and diets, yet our understanding of this special organ is lacking” – Please rewrite this sentence.

In the introduction section: Authors should add information supported with appropriate references that endorse their hypothesis especially for the gene expression study.

Line 393: euthanized using buffered MS-222 – Add appropriate dose accompanied with a valid reference.

Line 396: 4% paraformaldehyde or 10% paraformaldehyde – Please confirm

Line 405: (Zhang et al., 2022) – This reference is not present in the reference list. You should add it and also use MDPI reference style for this citation.

Line 469: (Livak and Schmittgen, 2001) - This reference is not present in the reference list. You should add it and also use MDPI reference style for this citation

Line 469: Why you selected β-actin as a reference gene or housekeeping gene for your study? Is the use of single housekeeping gene being enough?

Supplementary Table S2:

This table should be presented in the manuscript text not in the supplementary material. This table is important for readers to replicate your results. Moreover, melting temperatures, NCBI GenBank accession numbers, primer efficiency %, and annealing temperatures should be presented in the table.

References: (Minor revisions)

Reference section contains several minor revisions in the Journal names.

Line 510: Marine Biotechnology

Line 581: Indian Journal of Fisheries.

Line 583: The Journal of Clinical Endocrinology and Metabolism.

Line 585: DNA Research

Line 590: The Journal of Cell Biology

Line 601: Biological Research.

Line 602: The FEBS Journal.

Line 607: Journal of Fish Biology

Line 610: DNA Research

Line 617: Seminars in Immunology.

Line 618: Ctenopharyngodon idella – Write italic

Line 619: Developmental and Comparative Immunology.

Reviewer 2 Report

Review for the paper "Morphological and molecular evidence for functional of the pharyngeal sac in the digestive tract of silver pomfret (Pampus argenteus)" by Huan Jiang, Jiabao Hu, Huihui Xie, Man Zhang, Chuanyang Guo, Youyi Zhang, Yaya Li, Cheng Zhang, Shanliang Xu, Danli Wang, Xiaojun Yan, Yajun Wang, Xubo Wang submitted to "International Journal of Molecular Sciences".

Stromateoidei is a relatively small suborder of Perciformes currently comprising 6 families, 16 genera, and about 65 species. They occur worldwide in coastal and oceanic waters of tropical to temperate regions. All of these fishes are well characterized by the presence of a pharyngeal sac, located between the pharynx and esophagus and containing many teeth. The pharyngeal sac is a saccular organ located on the proximal portion of the esophagus with internally muscular walls covered by numerous papillae. The pharyngeal sac has been studied for morphology and histology and the application of these characteristics for taxonomic purposes. Despite this, the gross anatomy and histology are not well described. Also, there are no data on the molecular properties of the pharyngeal sac. Therefore, the topic of this article is interesting and may be useful for a better understanding of the structure and role of the pharyngeal sac in stromateoidei fish. The authors studied this organ in the silver pomfret Pampus argenteus, a commercially important fish species. The authors provided a detailed description of the pharyngeal sac coupled with photographs and provided results of transcriptome analysis of this organ. There are some concerns to be fixed prior to the acceptance of this paper.

Major concerns.

Figures 8 and 9. The authors should increase the font size.

How many fish specimens were used in this study?

As the authors mentioned that "the role of pharyngeal sac mainly in shredding food and assisting in pre-digestion of food items", they should provide information about the diet of Pampus argenteus under natural conditions and relate that diet with the food items used in this study. Do the authors' results accurately reflect the role of the pharyngeal sac for natural populations or these may be applied to cultivated fish? To which degree?

The text contains a number of grammatical errors and misspelling and should be revised thoroughly.

Specific remarks.

L 10. Consider replacing " pharyngeal sac plays" with "the pharyngeal sac play.

L 14. Consider replacing " the well-developed muscular walls" with " well-developed muscular walls.

L 21. Consider replacing " These results supported that the role" with " These results supported the role.

L 23. Consider replacing " about the morphological structure of pharyngeal sac, supported the adaptation in" with " of the morphological structure of pharyngeal sac and supported the adaptation in the"

L 24. Consider replacing " digestive tract for feeding gelatinous prey" with " the digestive tract for feeding gelatinous prey".

L 24. Consider replacing " a first report" with " the first report" .

L 25. Consider replacing " and served a reference" with " providing a reference" .

L 31. Consider replacing " assigned with a classic role " with " assigned the classic role " .

L 36. Consider replacing " and have the unusual diet" with " and have an unusual diet" .

L 40. Consider replacing " morphological characteristic can" with " morphological characteristics that can" .

L 46. Consider replacing " the  related molecular research about" with " related molecular research on" .

L 51. Consider replacing " has been studied for a long time [19]. However, due to lack" with " have been studied for a long time [19]. However, due to a lack" .

L 53. Consider replacing " It known to that the structural" with " It is known that the structural" .

L 57. Consider replacing " digestive tract in many  marine and freshwater fishes has" with " the digestive tract in many  marine and freshwater fishes have" .

L 76. Consider replacing " with mean size 6.3 ± 0.27 cm and mean length" with " with a mean size of 6.3 ± 0.27 cm and a mean length of" .

L 81. Consider replacing " The papilla had an elongate" with " The papilla had an elongated" .

L 92. Consider replacing " elongate axis" with " elongated axis" .

L 111. Consider replacing " in bright area, I band and Z band in dark area" with " in the bright area, I band and Z band in the dark area" .

L 124. Consider replacing " Sarcomere appear as distinct structural  units: a band in bright area, I band in dark area" with " Sarcomeres appear as distinct structural  units: a band in the bright area, I band in the dark area" .

L 131. Consider replacing " of epithelium layer," with " of the epithelium layer," .

L 141. Consider replacing " from adipose tissue layer" with " from the adipose tissue layer" .

L 145. Consider replacing " For the second regions" with " For the second region" .

L 149. Consider replacing " exposed in the sac cavity with" with " exposed to the sac cavity with the" .

L 152. Consider replacing " that internal area of stellate base" with " that the internal area of the stellate base" .

L 154. Consider replacing " we drawn" with " we drew" .

L 164. Consider replacing " exposed  in" with " exposed  to" .

L 172. Consider replacing " showing the elongate axis" with " showing the elongated axis" .

L 176. Consider replacing " fractured of " with " fractured" .

L 193. Consider replacing " the difference between two groups" with " the difference between the two groups" .

L 226. Consider replacing " showed that there were large" with " showed that there was a large" .

L 270. Consider replacing " that the RT-qPCR data" with " that the RT-qPCR data were" .

L 276. Consider replacing " a clear description about the morphological" with " a clear description of the morphological" .

L 279. Consider replacing " layers included" with " layers including" .

L 280. Consider replacing " this study had noticed" with " this study noticed" .

L 286. Consider replacing " consisted  of  epithelium layer" with " consisted  of  the epithelium layer" .

L 289-291. Please, re-phrase.

L 298. Consider replacing " Similar papilla has been reported" with "A similar papilla has been reported" .

L 305. Consider replacing " jellyfish and shredding it" with " jellyfish and shredding" .

L 307. Consider replacing " may reflected the adaptation" with " may reflect the adaptation" .

L 308. Consider replacing " of these specialized digestive" with " of this specialized digestive" .

L 310. Consider replacing " papillae is interest" with " papillae is of interest" .

L 311. Consider replacing " the prey selection" with " prey selection" .

L 317. Consider replacing " few studies had been reported about the abundant adipocytes appeared  in the structure of digestive tract. It had" with " few studies have been reported about the abundant adipocytes appeared  in the structure of the digestive tract. It has" .

L 320. Consider replacing " to have a intradermal canal plexus lied" with " to have an intradermal canal plexus lying" .

L 322. Consider replacing " a recently report" with " a recent report" .

L 323. Consider replacing " contribute to protect" with " contribute to protecting" .

L 335. Consider replacing " genes determined from" with " genes determined from the" .

L 343. Consider replacing " reported that KRT8 expressed" with " reported that KRT8 is expressed" .

L 344. Consider replacing " KRT13 expressed" with " KRT13 is expressed" .

L 352. Consider replacing " To some degrees" with " To some degree" .

L 363. Consider replacing " inferred that the digestive" with " inferring that the digestive" .

L 364. Consider replacing " might stronger " with " might be stronger " .

L 366. Consider replacing " function of digestion" with "a function of digestion of" .

L 376. Consider replacing " whether pharyngeal sac have" with " whether the pharyngeal sac has" .

L 377. Consider replacing " description about" with " description of" .

L 398. Consider replacing " part were placed" with " part was placed" .

L 403. Consider replacing " to analysis" with " to analyze" .

L 406. Consider replacing " more than 48 h" with " for more than 48 h" .

L 408. Consider replacing " After stained" with " After being stained" .

L 410. Consider replacing " and storing in" with " and storing at" .

L 411. Consider replacing " positive fluorescence microscope" with " a positive fluorescence microscope" .

L 421. Consider replacing " with same buffer" with " with the same buffer" .

Round 2

Reviewer 1 Report

The authors have properly addressed the comments raised by the anonymous reviewer and manuscript merits acceptance in its current form

Author Response

Dear Editor,

         Thank you again for your positive comments on our manuscript.

All the best with your work and have a nice day.

Kind regards,

Xubo Wang

Reviewer 2 Report

Second Review for the paper "Morphological and molecular evidence for functional of the pharyngeal sac in the digestive tract of silver pomfret (Pampus argenteus)" by Huan Jiang, Jiabao Hu, Huihui Xie, Man Zhang, Chuanyang Guo, Youyi Zhang, Yaya Li, Cheng Zhang, Shanliang Xu, Danli Wang, Xiaojun Yan, Yajun Wang, Xubo Wang submitted to "International Journal of Molecular Sciences".

The authors have revised the paper well. I have only one minor concern:

P. 7-8. Figures 5–7 are, in fact, present as a sole combined figure. Therefore, these figures should be separated or the figure caption and citations in the text should be modified.
